# Wideband SiGe-HBT Low-Noise Amplifier with Resistive Feedback and Shunt Peaking

**DOI:** 10.3390/s23156745

**Published:** 2023-07-28

**Authors:** Ickhyun Song, Gyungtae Ryu, Seung Hwan Jung, John D. Cressler, Moon-Kyu Cho

**Affiliations:** 1Department of Electronic Engineering, Hanyang University, Seoul 04763, Republic of Korea; isong@hanyang.ac.kr; 2Division of Nanoscale Semiconductor Engineering, Hanyang University, Seoul 04763, Republic of Korea; daeyeon36@hanyang.ac.kr; 3GRIT Custom-IC Corp., Seoul 01886, Republic of Korea; sh.jung@gritcic.com; 4School of Electrical and Computer Engineering, Georgia Institute of Technology, Atlanta, GA 30318, USA; cressler@ece.gatech.edu; 5Department of Computer Engineering, Korea National University of Transportation, Chungju-si 27469, Republic of Korea

**Keywords:** cascode, inductive emitter degeneration, low-noise amplifier (LNA), resistive feedback, SiGe HBT, shunt peaking, wideband

## Abstract

In this work, the design of a wideband low-noise amplifier (LNA) using a resistive feedback network is proposed for potential multi-band sensing, communication, and radar applications. For achieving wide operational bandwidth and flat in-band characteristics simultaneously, the proposed LNA employs a variety of circuit design techniques, including a voltage–current (shunt–shunt) negative feedback configuration, inductive emitter degeneration, a main branch with an added cascode stage, and the shunt-peaking technique. The use of a feedback network and emitter degeneration provides broadened transfer characteristics for multi-octave coverage and a real impedance for input matching, respectively. In addition, the cascode stage pushes the band-limiting low-frequency pole, due to the Miller capacitance, to a higher frequency. Lastly, the shunt-peaking approach is optimized for the compensation of a gain reduction at higher frequency bands. The wideband LNA proposed in this study is fabricated using a commercial 0.13 μm silicon-germanium (SiGe) BiCMOS process, employing SiGe heterojunction bipolar transistors (HBTs) as the circuit’s core active elements in the main branch. The measurement results show an operational bandwidth of 2.0–29.2 GHz, a noise figure of 4.16 dB (below 26.5 GHz, which was the measurement limit), and a total power consumption of 23.1 mW under a supply voltage of 3.3 V. Regarding the nonlinearity associated with large-signal behavior, the proposed LNA exhibits an input 1-dB compression (IP1dB) point of −5.42 dBm at 12 GHz. These performance numbers confirm the strong viability of the proposed approach in comparison with other state-of-the-art designs.

## 1. Introduction

Recent advances in cell phones, WLAN, 5G/6G, and Internet of things (IoT) technologies place ever-increasing demands on the performance of wireless transceivers [1,2]. In addition, as more radio-frequency (RF) communication protocols and require the support of one-chip solutions, higher levels of integration have become mandatory for simpler system configuration and cost-effective implementation [3]. Therefore, RF designers should minimize the use of off-chip components, such as RF chokes and capacitors, and optimize the system performance for wider operational bandwidth in the form of radio-frequency integrated circuits (RFICs). Ideally, component circuit blocks in RF front-ends support flat in-band characteristics, with low fluctuations (or variations) in performance numbers. In this way, additional compensation circuitry or signal processing logic may be unnecessary, or they can be simplified in order to minimize the burden on full systems.

In terms of RF receiver operation, the overall system performance is largely affected by a low-noise amplifier (LNA). Since it is the first gain stage of the entire receiver chain, it dominates the noise figure (NF) of a receiver and plays a key role in matching the characteristics between an antenna and a front-end chipset. The power gain of an LNA is one of the important design factors used to sufficiently amplify a weak input signal from an antenna for subsequent processing. Since the relationship between the power gain and linearity has a trade-off, however, the power gain should be properly balanced under the consideration of the linearity of the entire system. On the other hand, when the power gain increases, the noise characteristics are typically improved at the cost of power consumption. As the achievable minimum noise figure is limited by many factors, including specific technologies (CMOS, III–V, and etc.), circuit topology, types of devices, and DC-biasing conditions, designers should carefully choose and optimize the architectures and the parameters of an LNA, analyzing the system performance/power budgets and requirements. Thus, the optimal design of an LNA should consider all performance aspects: high gain/linearity, low noise figure, power dissipation, and good input/output impedance matching [4]. For extending the bandwidth of an LNA for a wideband system, several approaches, such as distributed topology, feedforward/feedback network, or shunt peaking, have been suggested in the literature. Depending on the specifications or the system requirement, it is critical to select and combine the proper design techniques, maximizing the advantages of each approach and addressing or minimizing the associated potential drawbacks.

When it comes to the realization of a wideband LNA, there are many available choices for the implementation of semiconductor technologies. In general, the semiconductor technologies can be chosen based on the important requirements and specifications of a system. For example, if power handling capability is more important, III-V technologies such as GaN might be considered. For an integrated system-on-a-chip design, however, the weaknesses such as high fabrication cost, small wafer size (yield), and difficulties of integration with other digital processing circuits make III-V technologies less ideal. To overcome these drawbacks, silicon-based technologies (CMOS) could be an option, but the large parasitic capacitances associated with MOSFETs would lead to design challenges in regards to covering multiple frequency bands.

Considering high-frequency characteristics, CMOS compatibility, use of existing intellectual properties (IP), and tape-out cost, silicon–germanium (SiGe) BiCMOS technology is one of the suitable choices [5]. The SiGe heterojunction bipolar transistors (HBTs) in this platform provide low device parasitics, high unity-gain frequencies (*f_T_*)/maximum oscillation frequencies (*f_MAX_*), low noise performance under reasonable DC power consumption, moderate breakdown voltages, and good device linearity. All these properties are essential for designing wideband LNAs [6,7]. Hence, SiGe-HBT-based LNA topologies have been investigated in this study [8,9]. As discussed later, the proposed circuit employs a negative resistive feedback loop, a cascode stage with an emitter-degeneration inductor, and the shunt-peaking technique to achieve wideband operation.

In this work, we propose a wideband SiGe-HBT LNA that utilizes the aforementioned design approaches and techniques. In Section 2, some of the conventional wideband LNA topologies are reviewed in terms of operation principles, advantages, and drawbacks. Then, Section 3 shows the design and the optimization of the proposed LNA with a detailed theoretical analysis of input impedance, signal gain, and noise figure, based on simplified small-signal equivalent circuits. Section 4 presents the simulation results and the measurement data, along with a discussion. In addition, a performance comparison with other state-of-the-art wideband LNA designs is provided for overall evaluation. Lastly, Section 5 summarizes the findings of this research.

## 2. Review of Wideband LNA Core Topology

For designing wideband LNAs, several approaches have been proposed and utilized in the literature. In general, the main topology of wideband LNAs is categorized into the common-base (CB) and the common-emitter (CE) stages [10,11]. Figure 1 shows the core structures of some of the conventional design techniques. First, Figure 1a represents a wideband LNA using a CB stage. The input signal is delivered to the emitter of Q_1_, and the output of the LNA is carried to the collector of Q_1_. At the input side, a passive component (Z_E_) is inserted between the emitter and the ground, and its purpose is to block the signal leakage by presenting a large impedance (ideally an AC open). The input signal simultaneously drives the feedforward amplifier, and the amplifier output controls the AC base voltage of the transistor. When a conventional CB structure, without an additional amplifier, is selected for the wideband design, the input matching is relatively straightforward, in contrast to the case of CE [12,13]. Since the effective impedance looking into the emitter terminal of the main transistor is equivalent to 1/g_m_, this impedance can be optimized to the standard system impedance of 50 Ohm by adjusting the transistor size and the bias [14]. Regarding the gain of a CB LNA, however, due to the tradeoff relation between the signal gain and the noise figure, it is difficult to simultaneously obtain a low noise figure and a high gain. In addition, the fact that from a simple CB stage, the lowest boundary of the achievable noise figure is 3 dB is another weakness. For decoupling both performance parameters against each other, a feedback or feedforward amplifier is often included for optimization, as shown in Figure 1a.

To mitigate the drawbacks of CB-based LNAs and achieve wide operating bandwidth, a CE stage can be utilized for wideband LNA design [15]. Figure 1b,c show examples of CE-based LNAs; the former illustrates an emitter degeneration with a series impedance at the base, and the latter presents a feedback topology between the base and the collector. The primary advantages of this approach include a generally lower noise figure than the CB counterpart, but the impedance matching requires additional circuitry. When the base (or the gate, in a MOSFET) being used is an input terminal, the impedance seen is mostly capacitive in the form of c_be_ or c_gs_, respectively [16]. In order to provide a relatively low real impedance to the input side, a degeneration inductor between the emitter and the ground is inserted, as illustrated in Figure 1b. To obtain wideband characteristics, a complex impedance network at the base of the input transistor is often implemented for shaping the target filter response. It is often composed of multiple lumped components (such as a π network) for low RF ranges [17] or distributed elements for millimeter wave applications. The disadvantage of this approach is large on-chip area due to the passive components. Because the base/emitter impedances are implemented with multiple inductors or capacitors, the overall chip tends to exhibit a higher loss as the operation frequency increases, and the required chip area of inductors or transmission lines can be excessively large [18].

As an alternative, a negative feedback loop is adopted between the input and the output of an LNA [8,17,19,20] (see Figure 1c). Similar to the previous case, the input signal drives the base terminal of Q_1_, whereas the output is at the collector terminal. The main feature of this schematic is that it utilizes a feedback loop where Z_F_ connects two terminals, as shown in Figure 1c. Depending on the type of impedance (real or imaginary) on the feedback path, it is divided into either resistive or reactive feedback, respectively. By analyzing the open-loop and the feedback factor of the circuit, the input impedance can be matched to the source. With the help of the feedback properties, the overall frequency response becomes smoother and wider, and the circuit exhibits improved linearity due to a lower closed-loop gain. In spite of these benefits, the feedback network does not operate well at higher frequencies because of the reduced open-loop gain itself and the effect of parasitics associated with active/passive devices.

Based on the wideband-design techniques above, we have investigated the pros and cons of each approach and combined them, maximizing the benefits of each method. For instance, in order to simultaneously optimize gain and noise performance, a CE-based topology exhibits advantages over a CB stage. In addition, because a well-defined real input impedance is required for matching without degrading other metrics, an emitter degeneration must be included. Next, the goal of wideband transfer characteristics can be achieved with a negative feedback. When the target bandwidth is in multi-octave ranges, the use of a feedback loop only does not provide enough coverage, especially at a higher corner frequency. In this case, design approaches such as shunt-peaking or a cascade stage may be merged into the design. Details of the proposed wideband LNA will be discussed in more detail in the next section.

## 3. Design of the Proposed SiGe-HBT LNA

As mentioned in the previous section, a variety of wideband approaches have been utilized. The schematic of the proposed wideband SiGe-HBT LNA is shown in Figure 2. Its core includes a CE stage (Q_1_) and a CB stage (Q_2_), in a typical cascode configuration. The CB transistor suppresses the Miller effect by lowering the voltage gain from the base of Q_1_ to the emitter of Q_2_. This also enhances the wideband operation of the LNA by pushing the corresponding input pole to a higher frequency [16,20]. Another advantage of a cascode configuration includes the increased output impedance. When a test signal is looking into the collector of Q_2_, the output impedance of an LNA is boosted by a factor of the product of the transconductance and the output resistance of Q_2_. With the increased output impedance, the overall voltage gain increases [17]. As long as the operation frequencies are lower than the unity-gain frequency, the noise contribution from the CB stage is much less dominant in the CE stage. 

For generating a resistive impedance at the input interface of the LNA, a degeneration inductor (L_E_) is inserted between the emitter of Q_1_ and the ground. Along with the series inductor (L_B_) at the base of Q_1_, it can present a reasonable input impedance, noise matching, and moderate linearity [16]. With its high quality factor (Q-factor) of L_E_, this technique has been widely adopted in narrowband LNAs, but it becomes useful in a wideband LNA if a negative feedback is employed around the LNA. Since a negative feedback network broadens the overall Q-factor (de-Qing), the input obtains a real impedance for a wide range of frequencies. As a result, it is possible to achieve impedance matching and optimized NF over those of the target operation bands. In the aspect of implementation, a degeneration inductor is often realized by a custom-designed spiral metal structure or a simple transmission line, due to its small required inductance. After the L_E_ inductance and the feedback impedance have been fixed, the base inductor (L_B_) needs to be optimized to support and maintain wideband characteristics.

In the proposed wideband LNA, the core stage is configured with a negative feedback loop via a resistor and a capacitor (Figure 2). The use of feedback resolves the tradeoff relationship between NF and input matching [8]. The selection of a feedback resistance (R_F_) and a feedback capacitance (C_F_) should be performed carefully, since it affects the LNA performance when the close-loop is formed. In this design, the feedback operation is mostly determined by R_F_, whereas the C_F_ provides 1) DC-blocking between the collector of Q_2_ and the base of Q_1_ and 2) slight reactance turning for bandwidth optimization. The R_F_ needs to be within a reasonable range such that it maximizes the gain and the bandwidth, but does not overload the open-loop impedances. In addition to the general properties of feedback, it also exhibits the supplementary advantages of improved linearity and the desensitization of parameter variations within the circuit [19,20].

The shunt-peaking technique is adopted to broaden the bandwidth of the gain of the LNA. Conventionally, this approach is employed in analog amplifiers, but it can also be a useful approach in the design of a wideband LNA. By connecting an inductor in series with the resistor in the place of a load, one additional pole and one new zero are introduced into the impedance network [17]. These points collectively create a peak in the magnitude of the transfer function, which allows it to maintain an increased gain over a wider range of frequencies. By optimizing the shunt-peaking network, the gain reduction at the high-frequency range is compensated, and more flat in-band transfer characteristics can be obtained.

### Performance Analysis of the Proposed Wideband LNA

The proposed circuit can be analyzed by deriving the input impedance, the gain, and the noise figure. The simplified small-signal equivalent circuit is shown in Figure 3a. For a SiGe HBT, the Early effect is neglected, and the small-signal model includes only the base-to-emitter capacitor (C_BE_) and the transconductance (g_m_). The base-to-emitter resistance (r_π_) is ignored because its impedance is sufficiently larger than the capacitor impedance 1/sC_BE_ at the operating frequency. In addition, since the cascode device could be regarded as a current buffer, it is omitted in the analysis.

The equations derived in this section are obtained from an approximate method based on the shunt–shunt feedback configurations [21]. Only dominant circuit parameters are included in the small-signal circuit, whereas many other parasitics of transistors and inductors were ignored for manageable derivations. Another aspect of the approximate method is that the circuit in Figure 3a is assumed to operate as the standard feedback-system model: (1) the input signal flows from v_X_ to v_out_ only through the feedforward path (i.e., from the base to the collector of the input transistor), and (2) the output signal is returned back to v_X_ via the feedback resistor R_F_ only. These assumptions are not always valid when an LNA operates at high frequencies. The neglected parasitics will affect the overall response of the LNA, and the reduced open-loop gain limits the bandwidth extension. At low frequencies or center frequency, however, the feedback mechanism operates as it is intended, with the open-loop (transimpedance) gain of equal to or greater than 20–30 dB. Thus, it is possible to apply the feedback theory for analysis. The frequency response obtained from the approximate method may be different from that of the exact analysis due to the use of a feedback system model. On the other hand, if we consider the complexity of the calculation from the exact analysis, the feedback equations provide additional insight into the LNA design and the optimization process, since they present the definitions of important terms in the equations, such as the open-/closed-loop impedances and gain.

After breaking the loop by separating R_F_ into two resistors (Figure 3b), the open-loop parameters can be derived. The open-loop input impedance looking into the base of Q_1_ is derived from the following equations. The expressions of the base-to-emitter voltage (v_BE_) and the emitter voltage (v_E_) are shown in Equations (1) and (2), respectively.
(1)$$v_{BE}=i_{B}\frac{1}{{sc}_{BE}}$$
(2)$$v_{E}=v_{B}-v_{BE}=v_{B}-i_{B}\frac{1}{{sc}_{BE}}$$

At v_E_, by applying KCL, we have the following results;
$$i_{B}+g_{m}v_{BE}=\frac{v_{E}}{{sL}_{E}}\to i_{B}+g_{m}i_{B}\frac{1}{{sc}_{BE}}=\frac{v_{B}}{{sL}_{E}}-\frac{i_{B}}{s^{2}L_{E}c_{BE}}$$
(3)$$i_{B}\left(1+\frac{g_{m}}{{sc}_{BE}}+\frac{1}{s^{2}L_{E}c_{BE}}\right)=\frac{v_{B}}{{sL}_{E}}$$

The open-loop impedance with the shunt R_F_ (Z_X,open_) is a parallel connection of R_F_ and Z_B,_ as shown below.
(4)$$Z_{B,open}=\frac{v_{B}}{i_{B}}={sL}_{E}\left(1+\frac{g_{m}}{{sc}_{BE}}+\frac{1}{s^{2}L_{E}c_{BE}}\right)$$
(5)$$Z_{X,open}=R_{F}||z_{B}=R_{F}||{sL}_{E}\left(1+\frac{g_{m}}{{sc}_{BE}}+\frac{1}{s^{2}L_{E}c_{BE}}\right)$$

Then, the transimpedance gain (R_o,open_) can be derived, using the voltage gain from v_X_ to v_out_. First, we see that the output impedance (Z_O_) is a parallel combination of R_F_ and Z_L_.
(6)$$Z_{O}=R_{F}||z_{L}$$

By applying KCL at v_E_, Equation (7) is obtained.
$$\frac{v_{X}-v_{E}}{1/{sc}_{BE}}+g_{m}(v_{X}-v_{E})=\frac{v_{E}}{{sL}_{E}}$$
(7)$$v_{E}=\frac{g_{m}+{sc}_{BE}}{\left(\frac{1}{{sL}_{E}}+{sC}_{BE}+g_{m}\right)}v_{X}$$

Now, the result of Equation (3) is used in the KCL equation at the node of v_out_.
$$\frac{v_{out}}{Z_{O}}+g_{m}(v_{X}-v_{E})=0$$
(8)$$\frac{v_{out}}{Z_{O}}+g_{m}\left[1-\frac{g_{m}+{sc}_{BE}}{\left(\frac{1}{{sL}_{E}}+{sC}_{BE}+g_{m}\right)}\right]v_{X}=0$$

By rearranging Equation (8), the voltage gain from v_X_ to v_out_ is,
(9)$$\frac{v_{out}}{v_{X}}=\frac{-g_{m}Z_{O}}{{sL}_{E}\left(\frac{1}{{sL}_{E}}+{sc}_{BE}+g_{m}\right)}$$

The open-loop transimpedance gain (R_o,open_) from the input current (i_in,open_) to v_out_ is derived as follows, using Equations (5) and (9).
(10)$$R_{o,open}=\frac{v_{out}}{i_{in,open}}=\frac{v_{out}}{v_{X,open}/Z_{X,open}}=z_{X,open}\frac{g_{m}Z_{O}}{{sL}_{E}\left(\frac{1}{{sL}_{E}}+{sC}_{BE}+g_{m}\right)}$$

For obtaining the closed-loop input impedance at v_in_, it is necessary to determine the feedback factor (K_f_). The voltage-to-current gain from v_out_ to the node X (K_f_) is equivalent to 1/R_F_, and the corresponding loop gain is,
(11)$$K_{f}R_{o,open}=\frac{1}{R_{F}}\frac{g_{m}Z_{O}Z_{X,open}}{{sL}_{E}\left(\frac{1}{{sL}_{E}}+{sC}_{BE}+g_{m}\right)}$$

The closed-loop input impedance z_X,closed_ and the overall input impedance z_in,closed_ are shown in Equations (12) and (13), respectively. The input impedance (Z_in,closed_) of the LNA is the sum of sL_B_ and Z_X,closed_, and the parameters must be optimized to deliver 50 ohm to the driving stage.
(12)$$Z_{X,closed}=\frac{Z_{X,open}}{1+K_{f}R_{o,open}}=\frac{R_{F}||{sL}_{E}\left(1+\frac{g_{m}}{{sc}_{BE}}+\frac{1}{s^{2}L_{E}c_{BE}}\right)}{1+\frac{1}{R_{F}}\frac{g_{m}Z_{O}Z_{X,open}}{{sL}_{E}\left(\frac{1}{{sL}_{E}}+{sC}_{BE}+g_{m}\right)}}$$
(13)$$Z_{in,closed}=sL_{B}+\frac{R_{F}||\frac{{sL}_{E}}{{sc}_{BE}}\left(\frac{1}{{sL}_{E}}+{sC}_{BE}+g_{m}\right)}{1+\frac{1}{R_{F}}\frac{g_{m}Z_{O}Z_{X,open}}{{sL}_{E}\left(\frac{1}{{sL}_{E}}+{sC}_{BE}+g_{m}\right)}}$$

The voltage gain (A_v_ = V_out_/V_in_) of the proposed LNA can be derived by using the closed-loop equations above. Under the voltage-driven condition, the input current is converted from the input impedance (Z_in,closed_), and the closed-loop transimpedance gain is then multiplied to generate the output voltage.
(14)$$R_{o,closed}=\frac{R_{o,open}}{1+K_{f}R_{o,open}}=\frac{\frac{g_{m}Z_{O}Z_{X,open}}{{sL}_{E}\left(\frac{1}{{sL}_{E}}+{sC}_{BE}+g_{m}\right)}}{1+\frac{1}{R_{F}}\frac{g_{m}Z_{O}Z_{X,open}}{{sL}_{E}\left(\frac{1}{{sL}_{E}}+{sC}_{BE}+g_{m}\right)}}$$
$$A_{v}=\frac{v_{out}}{v_{in}}=\frac{v_{out}}{Z_{in,closed}i_{in,closed}}=\frac{1}{Z_{in,closed}}R_{o,closed}$$
$$=\frac{1}{sL_{B}+\frac{R_{F}||\frac{{sL}_{E}}{{sc}_{BE}}\left(\frac{1}{{sL}_{E}}+{sC}_{BE}+g_{m}\right)}{1+\frac{1}{R_{F}}\frac{g_{m}\left(R_{F}||z_{L}\right)\left[R_{F}||{sL}_{E}\left(1+\frac{g_{m}}{{sc}_{BE}}+\frac{1}{s^{2}L_{E}c_{BE}}\right)\right]}{{sL}_{E}\left(\frac{1}{{sL}_{E}}+{sC}_{BE}+g_{m}\right)}}}\times$$
(15)$$\frac{\frac{g_{m}Z_{O}\left[R_{F}||{sL}_{E}\left(1+\frac{g_{m}}{{sc}_{BE}}+\frac{1}{s^{2}L_{E}c_{BE}}\right)\right]}{{sL}_{E}\left(\frac{1}{{sL}_{E}}+{sC}_{BE}+g_{m}\right)}}{1+\frac{1}{R_{F}}\frac{g_{m}\left(R_{F}||z_{L}\right)\left[R_{F}||{sL}_{E}\left(1+\frac{g_{m}}{{sc}_{BE}}+\frac{1}{s^{2}L_{E}c_{BE}}\right)\right]}{{sL}_{E}\left(\frac{1}{{sL}_{E}}+{sC}_{BE}+g_{m}\right)}}$$

For the optimization of the bandwidth in-band characteristics, the load impedance (Z_O_) should be properly designed. To maximize the effect of the shunt-peaking technique, the location of the associated zero is set around the upper corner frequency, whereas the introduced pole appears at a higher frequency than that of the bandwidth.

The simplified small-signal equivalent circuit with noise sources is shown in Figure 4. Among the many noise sources in the LNA, the dominant sources are included in the analysis. The input transistor Q_1_ exhibits its shot noise at the emitter-base junction and at the collector-base junction, whereas the feedback resistor and the base inductor present thermal noise. Using the Z_in_ and A_v_ equations above, the NF of an LNA is calculated as shown below.
(16)$$NF=1+\frac{\bar{V_{n,LNA}^{2}}}{4kTR_{S}{\left|\alpha \right|}^{2}A_{v}^{2}},$$

In (6), $\bar{V_{n,LNA}^{2}}$, k, T, R_S_, and α are the noise contributions of the LNA, the Boltzmann constant, the absolute temperature, the source impedance, and the impedance division between the source and the amplifier, respectively. Under an input-matched condition, the simplified $\bar{V_{n,LNA}^{2}}$ is derived as follows.
(17)$$\bar{V_{n,LNA}^{2}}=4kTA_{v}^{2}R_{LB}+\left\{\frac{4kT}{R_{F}}+2qI_{B}+\frac{1}{2}qI_{C}{\left(\frac{\omega C_{BE}}{g_{m}}\right)}^{2}\right\}\cdot {R_{X,closed}}^{2},$$
where R_X,closed_ is the close-loop transimpedance gain from the node X to the output shown in Figure 3a.
(18)$$R_{X,closed}=\frac{\frac{g_{m}Z_{O}z_{X,open}}{{sL}_{E}(\frac{1}{{sL}_{E}}+{sc}_{BE}+g_{m})}}{1+\frac{1}{R_{F}}\cdot \frac{g_{m}Z_{O}z_{X,open}}{{sL}_{E}(\frac{1}{{sL}_{E}}+{sc}_{BE}+g_{m})}}$$

Based on the design Equations (1)–(18), the performance of the proposed LNA can be optimized for matching, feedback operation, gain, and noise figure. Due to simplification in the analysis procedure, however, the accurate models and simulations are required for the fine-tuning of the LNA. In addition, parasitics associated with interconnection structures or other distributed elements should be extracted from electromagnetic tools and included in the design, since they may significantly affect the overall circuit behavior, especially at high frequencies.

## 4. Simulation and Measurement Results

The proposed wideband LNA was designed using a commercial 0.13 μm SiGe BiCMOS technology. The optimized final parameters for wideband operation and flat in-band transfer performance are listed in Table 1. The overall design process begins with the size and the bias of the core transistors under the given power budget. With proper transistor dimensions, other matching elements and passives can be optimized for balanced performance. The fabricated chip micrograph of the fabricated LNA circuit is shown in Figure 5a. The chip size of the core LNA, excluding measurement pads and input/output interconnecting lines, was 0.17 mm^2^ (0.36 mm × 0.48 mm). The measurement was conducted using a probe station, a vector network analyzer, a signal source, a spectrum analyzer, and a noise source in an on-chip characterization setup (Figure 5b). The fabricated chip was mounted on the chuck of the probe station, and AC/DC signals were connected via probe tips. Appropriate performance parameters, as well as proper test instruments, equipment, and software tools, were used. Regarding DC operation, the proposed wideband LNA has a current consumption of 7 mA from a 3.3 V supply voltage.

The small-signal performance parameters are shown in Figure 6. The simulated S_21_ scattering parameters (s-parameters) are very close to the measurement results (Figure 6a), showing minimized variations within the operation bandwidth. In Figure 6b, the input reflection coefficient, S_11_, was below −10 dB from 2 to 29 GHz, confirming the viability of the proposed approaches, whereas the output reflection coefficient, S_22_, was not as wide as that of S_11_, requiring the use of a buffer for better matching.

In Figure 7, the simulated and measured noise characteristics are plotted versus the frequency. The simulation predicted accurate results for low frequencies below 4 GHz, but due to the imperfect noise modeling of the design kits, errors grew as the frequency increases. Because the input matching characteristics were slightly down shifted, it is suspected that the optimum noise is different between the simulation and the hardware. The actual noise measurement was conducted up to 26.5 GHz, due to the limitations of the noise source used in the test. The average NF of the proposed LNA was around 3.7 dB across the operation bandwidth.

The large-signal performance of the proposed LNA is presented in Figure 8. The measured 1-dB compression point (IP1dB) was referred to the input side at different operation frequencies. The results showed IP1dB numbers of −8.26, −5.42, and −5.60 dBm at 2, 12, and 24 GHz, respectively. In addition, from the circuit simulation, the input third-order point (IIP3) was estimated to be 1.32 dBm at 12 GHz.

Table 2 summarizes the performance of the proposed wideband SiGe-HBT LNA and compares it with other state-of-the-art LNAs presented in the literature. The proposed LNA achieves a wide operation bandwidth with a low NF, low power consumption, and good linearity among the state-of-the-art designs. Specifically, the proposed LNA exhibits strength regarding the fractional bandwidth and the noise figure, whereas the power dissipation and linearity are similar to the average performance of other designs. In the last column of Table 2, a sample narrowband LNA is included. In contrast to wideband LNAs, a narrowband LNA has advantages in regards to noise figure and power consumption, as well as linearity. For a better evaluation of the relative performance, figure-of-merit (FoM) equations are used. FoM is a combined function of fractional bandwidth, center frequency, gain, NF, power consumption, etc., and FoM_I_ is defined as follows [22].
(19)$${FoM}_{I}=\frac{S_{21(dB)}\cdot {BW}_{\left(\%\right)}\cdot f_{c(GHz)}}{({NF}_{\left(linear\right)}-1)\cdot P_{dc(mW)}}$$

The second FoM equation (FoM_II_) includes a large-signal parameter (IP1dB) to also address the linearity performance of LNAs [16].
(20)$${FoM}_{II}=\frac{S_{21(dB)}\cdot {BW}_{\left(\%\right)}\cdot {IP}_{1dB(mW)}}{\left({NF}_{\left(linear\right)}-1\right)}$$

As inferred from the comparison, the proposed LNA exhibits the highest numbers among all the other designs, demonstrating that the concept suggested in this work is effective in designing a high-performance wideband LNA.

## 5. Conclusions

The design and the optimization of a wideband SiGe-HBT LNA are presented. The proposed LNA uses multiple approaches for extending the bandwidth of the circuit, such as a resistive feedback network, an emitter-degeneration inductor, a cascode stage, and the shunt-peaking technique. In addition, a detailed analysis of matching, gain, and noise figure is provided, using small-signal models and equivalent circuits. The proposed LNA is implemented in a 0.13 μm SiGe-BiCMOS process, and the measurement shows a wide bandwidth of 2.0–29.2 GHz, simultaneously covering many frequency bands for communication and sensor applications. The overall performance confirms that the proposed LNA is a viable solution for the realization of wideband RF systems.

## Figures and Tables

**Figure 1 sensors-23-06745-f001:**
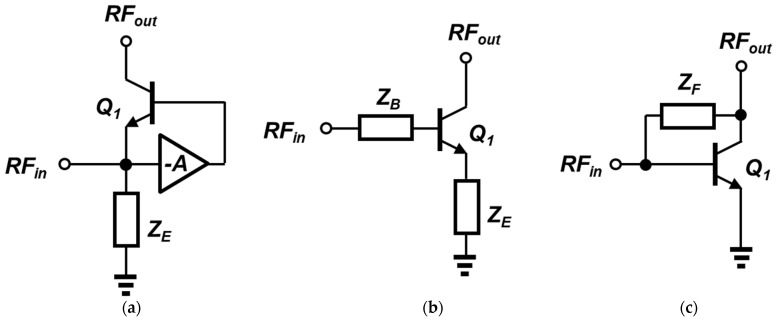
Conventional structures of wideband LNAs: (**a**) common-base stage with a feedforward amplifier; (**b**) common-emitter stage with a filter network; (**c**) common-emitter stage with a negative feedback loop.

**Figure 2 sensors-23-06745-f002:**
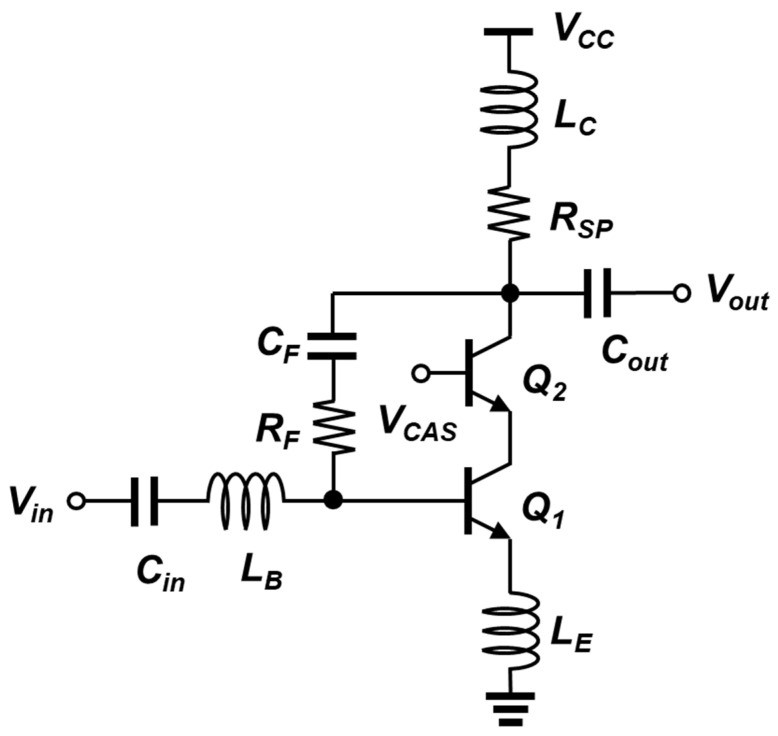
Schematic of the proposed wideband SiGe-HBT LNA.

**Figure 3 sensors-23-06745-f003:**
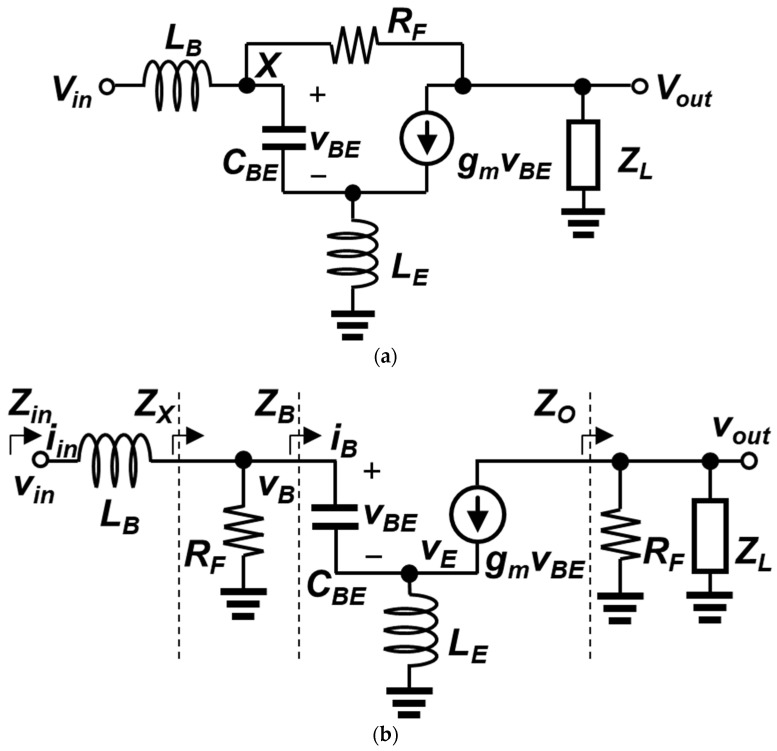
(**a**) Simplified small-signal equivalent circuit of the proposed LNA; (**b**) small-signal equivalent circuit with a broken loop for open-loop analysis.

**Figure 4 sensors-23-06745-f004:**
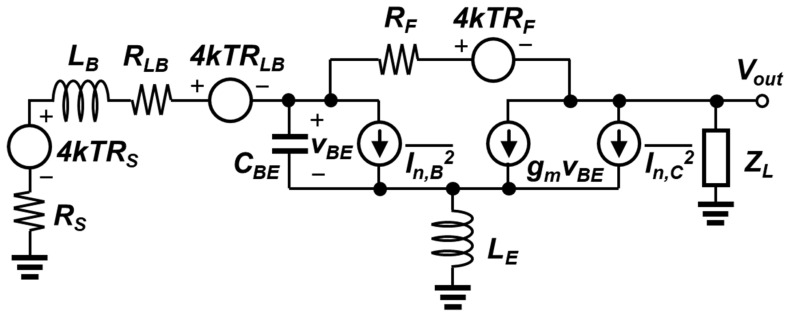
Simplified small-signal noise-equivalent circuit used for analysis.

**Figure 5 sensors-23-06745-f005:**
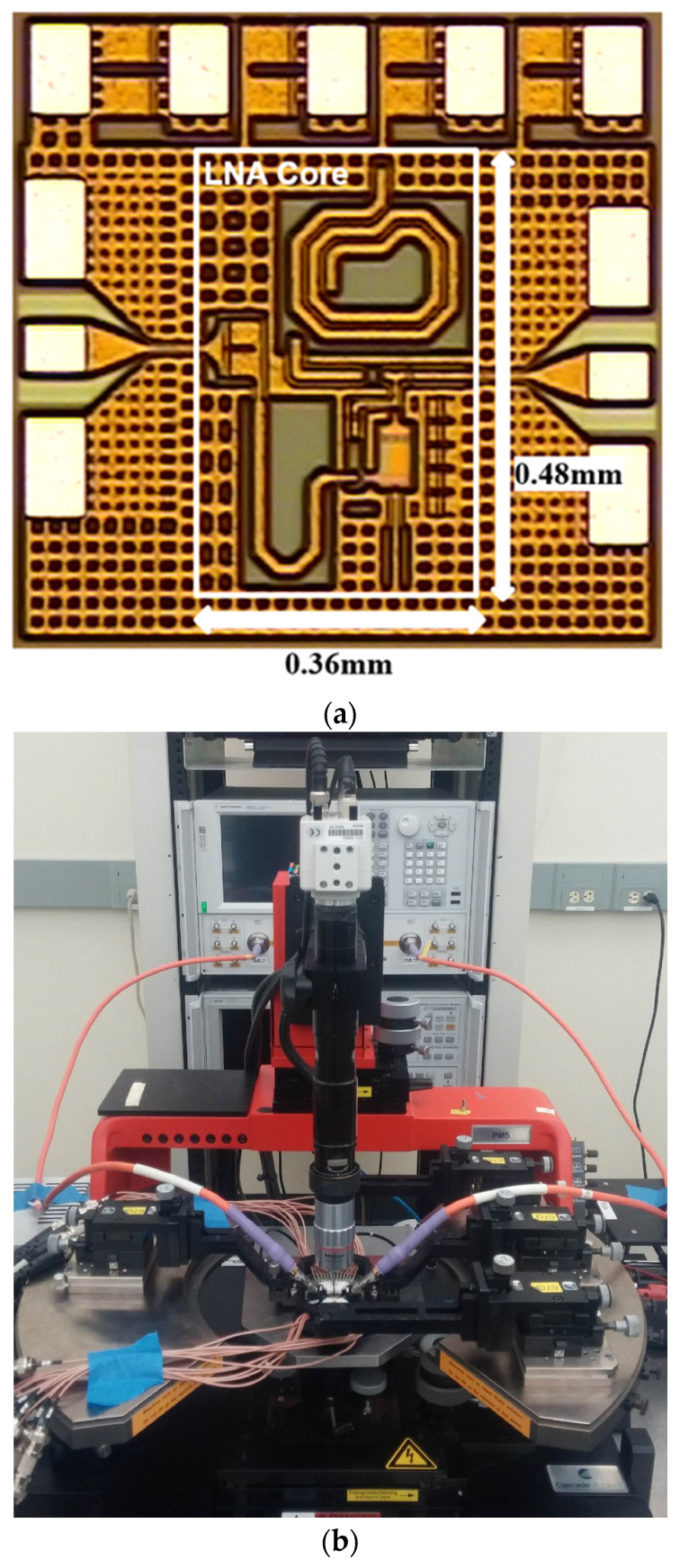
(**a**) Chip micrograph of the proposed wideband SiGe-HBT LNA. The core area of the LNA consumes 0.17 mm^2^. (**b**) On-wafer measurement setup.

**Figure 6 sensors-23-06745-f006:**
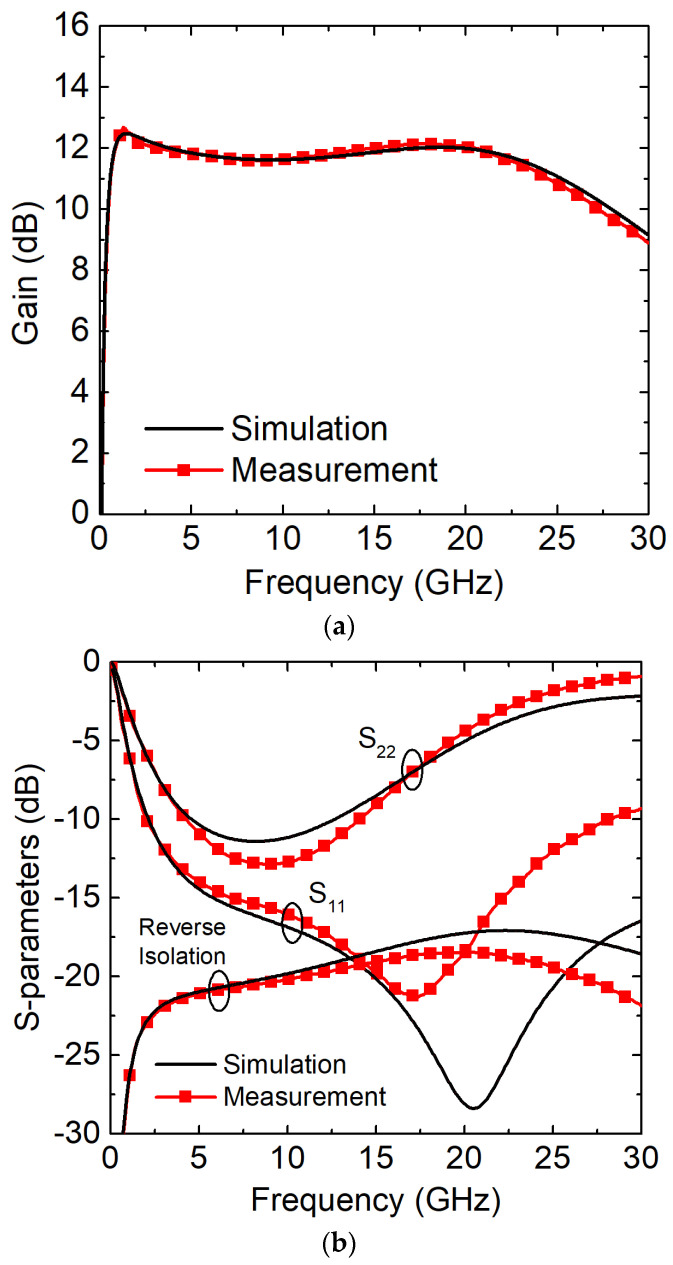
Simulated and measured (**a**) scattering parameters (s-parameters); (**b**) S_11_, S_22_, and S_12_ (reverse isolation) versus frequency.

**Figure 7 sensors-23-06745-f007:**
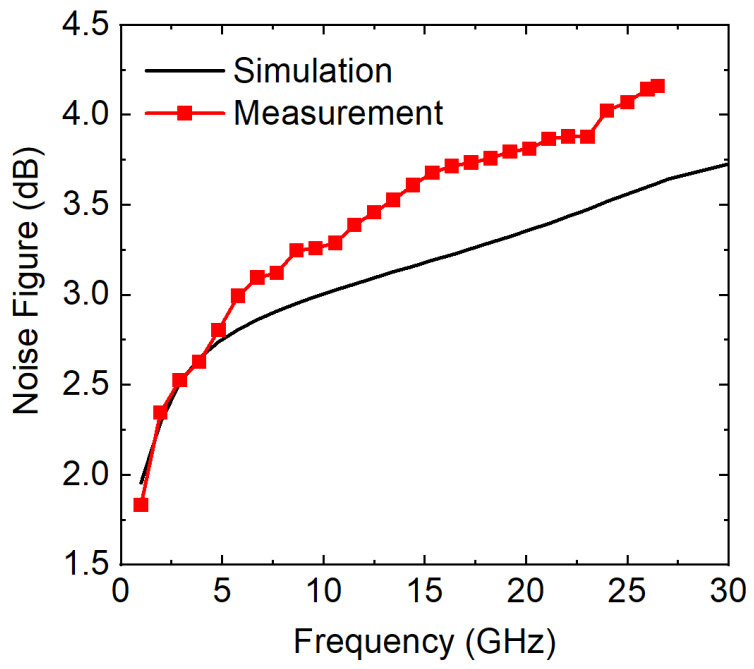
Simulated and measured noise figure (NF) versus frequency. The measurement was conducted up to 26.5 GHz.

**Figure 8 sensors-23-06745-f008:**
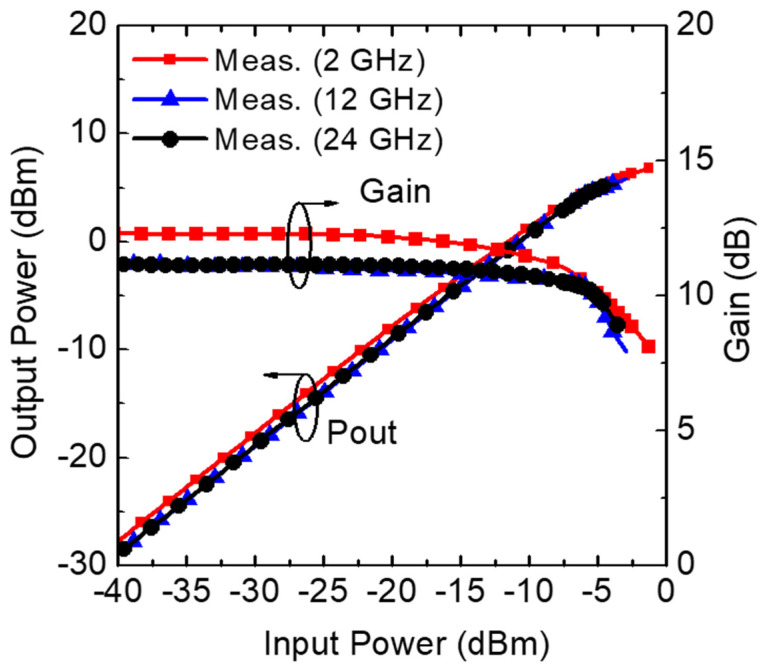
Measured input 1-dB compression point (IP1dB) of the proposed LNA.

**Table 1 sensors-23-06745-t001:** Design Parameters of LNA Components.

Component	Size/Value	Component	Size/Value
L_C_	900 pH	C_out_	2.2 pF
L_B_	150 pH	R_SP_	69 Ω
L_E_	20 pH	R_F_	240 Ω
C_F_	360 fF	Q_1_	3 × (0.12 × 8) μm^2^
C_in_	2 × 3.1 pF	Q_2_	3 × (0.12 × 8) μm^2^

**Table 2 sensors-23-06745-t002:** Comparison of LNA performance with state-of-the-art designs.

Spec.	This Work	[17]	[23]	[24]	[25]	[26]	[27]	[28]
Technology (nm)	SiGe 130	SiGe 180	SiGe	CMOS 180	CMOS 180	CMOS 90	CMOS 40	SiGe 130
Vcc (V)	3.3	3.3	3.6	1.8	N/A	1.2	1.2	4.8
S_21_ (dB)	9.7–12.4	16.1–18.1	16–18	11.6–12.6	10.6–11.8	8.5–10.7	14–17	17.5–20.5
S_11_ (dB)	<−10	<−10	<−9	<−8.6	<−9.84	<−10	<−10	<−10
BW (GHz)	2–29.2	2–20	0.1–23	1.5–11.7	2–12	1.6–28	1–11	7.5–11.8
FBW (%)	174.4	163.6	198.3	154.5	142.9	178.4	166.7	47.8
* NF (dB)	3.6	3.7	5	4.24	3.36	3.66	4.2	1.65 (aver)
P_dc_ (mW)	23.1	55	54	10.34	22.7	21.6	9	100
IP_1dB_ (dBm)	−5.42	−11.7	−8 *	−22	−8.4	−9	−12	−0.75
^+^ Area (mm^2^)	0.462	0.435	0.184 *	0.536	0.447	0.139	0.061	0.72
FoM_I_	1132	442	353	753	444	991	1159	190
FoM_II_	481	150	261	7	209	182	110	1781

* Average number; ^+^ Chip area without measurement pads and buffers.

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
