# Peer review of "Wideband SiGe-HBT Low-Noise Amplifier with Resistive Feedback and Shunt Peaking"

_sensors, 2023, doi:10.3390/s23156745_

Round 1

Reviewer 1 Report

This paper provides a wideband SiGe-HBT LNA with resistive feedback and shunt peaking, which is interesting. An ultra-wide band LNA with low NF is very important for ultra-wideband receiver. This paper still has some problems: 

1) The measured bandwidth is from 2-24GHz in the context,while in the abstract, the measured bandwidth is from 2-29GHz. Please give the measured results from 24GHz-29GHz;

2)The physical graph of the LNA should be shown in the paper, as well as the testbed.

Author Response

Please see the attached response.

Reviewer 2 Report

A design approach to optimize the performances of a wide LNA in SiGe-HBT tecnology is presented. Design approaches and choices are discussed. The amplifier has been realized and tested.

Overall I find this to be a sensible work. However, there are a few aspects that should be improved.

I have an issue with the idea that Q1 and Q2 in Fig. 2 are "in series" (line 121). I would replace "Its core is..........input device (Q1)" with "Its core is a CE stage (Q1)and a CB stage (Q2) in a typical cascode configurazion." or something similar. 

While there are many references for the configurations 1b and 1c, this is not the case for the configuration in 1a. Moreover, if we interpred the node Vb in Fig. 1a as a bias voltage, this would mean that it is grounded in the small signal equivalent circuit, with the output of the auxiliary amplifier (-A) shorted as well. Please clarify how the figure (1a) should be interpreted. 

I am not familiar with the process used to obtain the closed loop gain. In particular, it is not clear to me in which sense Fig. 3b has anything to do with Fig. 3a. This should either be clarified (together with the entire process to obtain the closed loop gain) or a reference should be added where a discussion of this approacg can be found. 

As far as the English is concerned, it seems mostly fine to me, but the meaning of a few sentences in not very clear. For instance, I am not sure about the meaning of the sentence starting at line 89. Since the paper is not exceedingly long, may be the authors can try to use a few more sentences instead of a single and quite convoluted one whenever appropriate. 

Author Response

Please see the attached response.

Reviewer 3 Report

Introduction: The trade-off between gain, linearity and noise figure should be well explained.

It is preferable to introduce figures showing the difference between the cited approaches.

The Transistor selection criteria and topology description should be more explained. Its a key factor for LNA design. 

Line 114:

It is better to introduce a comparative table with different LNA designs and hilight the strong and weak performances.  

The justification of a new design should be improved since  recent commercialized LNAs are sophisticated enaugh. 

Lines 131 and 132:

please add a reference and describe state-of-art narrowband LNAs.

Line 230:

Comparison of the consumption of the proposed design to the state-of-art should be added.

Line 238: 

why is the gain constant in the whole frequency band?

reason for difference between simulated and measured noise figures

Is the noise figure still acceptable? compariosn to state of the art.

English is acceptable.

Author Response

Please see the attached response.

Round 2

Reviewer 2 Report

In my previous review, I had asked to clarify the method employed for deriving the gain expression in Eq. 5. The authors did not provide any real answer. In their reply they listed a number of papers in which feedback is used to modify the response of a circuit. The cited papers are relative to a variety of applications. For instance, the paper "A CMOS Resistive Feedback Differential Low-Noise Amplifier With Enhanced Loop Gain for Digital TV Tuner Applications" is relative to a feedback resistance added to a an amplifier with a low impedance output, while in the case of the circuit in the submitted paper, the output of the main amplifier has a very high impedance.

May be, I was not clear enough: there are many approaches to deal with feedback circuits that allow to extract the "exact" transfer function or an “approximate” transfer function starting from a transformed version of the circuit. 

For the "exact" methods, it should not be difficult to recover the paper or the book in which the approach is proposed and demonstrated. 

For approximate methods, one should clearly discuss under which conditions the approximation holds and, also in this case, provide a convincing explanation or a reference for the very specific case at hand.

Stating that "Figure 3b represents the open-loop small-signal circuit including loading effects of the feedback path under the broken-loop condition, which is necessary for calculating open-loop performance", as the authors have done in their answer to my question, unfortunately does not explain anything. They should explain how and why the closed loop response is obtained using the procedure followed in the paper, or provide a specific (and not a vaguely related) reference. They have done neither of these things.

Because of the lack of explanation, I have spent some time trying to make sense of the equations leading to the calculation of the voltage gain in EQ. 5. 

It can be easily verified by anyone that in some limit cases (for instance RF becoming vanishingly small or LE becoming extremely large) the frequency response in Eq. 5 disagrees with what can be trivially calculated from the circuit in Fig. 3a. Therefore, we must conclude that, from the point of view of circuit theory, the response in Eq. 5 IS NOT the response of the circuit in Fig. 3a. In some cases, it is possible that Eq. 5 is a sufficiently good approximation of the actual response, bud this must be demonstrated.

In trying to make some sense of Eq. 1 to Eq 5, I repeated myself the calculations and it appears that Eq. 2 is incorrect, since, assuming that using Fig. 3b makes sense, RF should appear in parallel to ZL at the numerator of the fraction. This error may or may not be the reason for the inconsistency I have verified in Eq. 5, but errors are present that need to be corrected. 

In other words, the procedure used to obtain Eq. 5 is not justified and, besides, obvious errors are present in the equations.

It is my view that without seriously addressing these problems, the paper cannot be published.

English is reasonably good. Please note that I am not a native English speaking person. 

Reviewer 3 Report

My comments have been considered.

Round 3

Reviewer 2 Report

In my second review I (once again) requested the authors to provide a reference for the feedback analysis approach they have been using.

I also pointed out mistakes in the derivation of the equations that the authors have acknowledged and, hopefully, corrected.

Notwithstanding the fact that the authors recognize, in their reply, that they are using an approximated method, which should make obvious that a theorical background for the approach that has been used must be provided, no actual reference has been given (once again).

There are indeed several possible approaches to the approximated analysis of a circuit in a shunt-shunt configuration and no reader can be expected to be familiar with all of them. In the interest of fairness toward the authors, I have taken upon myself to search for a possible reference that they seem to be unable to find or unwilling to provide.

The approach used by the authors is discussed in detail in the book:

“Analysis and Design of Analog Integrated Circuits – Fourth Edition, Paul Gray, Paul Hurst, Steve Lewis and Robert Meyer, John Wiley and Sons, Inc., 2001, ISBN 0-471-32168-0”

In particular, sub-chapter 8.5.1 specifically deals with the shunt-shunt configuration and provides the theoretical background needed to explain how the circuit in Fig. 3b is related to the circuit in Fig. 3a in the submitted paper. The same subchapter provides meaning to the network functions used in the paper to reach an estimation of the closed loop response.

Provided that the authors include a clear reference to the book I mentioned when introducing the circuit in Fig. 3b and inform the reader that he can find justification for the approach that is being followed in that book, I have no further objection to the publication of the manuscript.

English is reasonably good. Please note that I am not a native English speaking person. 
